# Power-over-Fiber LPIT for Voltage and Current Measurements in the Medium Voltage Distribution Networks

**DOI:** 10.3390/s21020547

**Published:** 2021-01-14

**Authors:** Fabio R. Bassan, Joao B. Rosolem, Claudio Floridia, Bruno N. Aires, Rodrigo Peres, Javier F. Aprea, Carlos Alexandre M. Nascimento, Fabiano Fruett

**Affiliations:** 1CPQD Research and Development Center in Telecommunications, Campinas 13086-902, Brazil; rosolem@cpqd.com.br (J.B.R.); floridia@cpqd.com.br (C.F.); baires@cpqd.com.br (B.N.A.); rperes@cpqd.com.br (R.P.); 2IMS Power Quality, Porto Alegre 91160-310, Brazil; javier.aprea@ims.ind.br; 3CEMIG, Belo Horizonte 30190-131, Brazil; caxandre@cemig.com.br; 4UNICAMP—Universidade Estadual de Campinas, Campinas 13083-970, Brazil; fabiano@unicamp.br

**Keywords:** LPIT, power-over-fiber, PoF, voltage sensor, current sensor, hybrid conductor, distribution network, optical fiber

## Abstract

In this work, we present the design, laboratory tests, and the field trial results of a power-over-fiber (PoF) low power instrument transformer (LPIT) for voltage and current measurements in the medium voltage distribution networks. The new proposed design of this power-over-fiber LPIT aims to overcome the drawbacks presented by the previous technologies, such as the continuous operation (measuring and data transmission) for a wide current range conducted in the medium voltage transmission lines, damage due to lightning strikes, accuracy dependency on vibration, position and temperatures. The LPIT attends the accuracy criteria of IEC 61869-10 and IEC 61869-11 in terms of current and voltage accuracy and it attends the practical criteria adopted by Utilities companies including voltage measurements without removing the coating of the covered conductors. The PoF based LPIT was developed to be applied at 11.9 kV, 13.8 kV, and 23.0 kV phase-to-phase nominal voltages, and in two current ranges 1.25–30 A and 37.5–900 A. The digital data transmission of current, voltage, and temperature from the sensing unit to the processing unit uses a special synchronism technique and it is performed by two 62.5 µm multimode fibers in 850 nm. The optical powering in 976 nm is also performed by one 62.5 µm multimode fiber from the processing unit to the sensor unit. We presented all details of the sensor design and its laboratory characterization in terms of accuracy and temperature correction. We also presented the results of field tests of the sensor made in two different conditions: in a standard distribution network and an experimental hybrid fiber/power distribution network. We believe that these studies aim to incorporate optical fiber and devices, digital technologies, communications systems in electrical systems driving their evolution.

## 1. Introduction

New power grids are incorporating digital technologies, communications systems, and standards that are driving the evolution of failure detection, metering, and control layers to meet the 21st-century challenges [1]. Smart Grid sensors are relevant elements in this evolution that provide real-time data and status of the grids for real-time monitoring, protection, and control of grid operations [2,3]. Low-power instrument transformers (LPIT) are a new generation of smart meters that can transmit digital voltage and current data directly from the conductors, thus preserving signal integrity [4,5,6]. Typical electric parameters measured by LPIT include voltage, current, power flow. LPIT for power-quality measurements (PQM) includes sags and swells, voltage and current harmonics, percent of total harmonic distortion, and total demand distortion. 

LPIT must attend to several stringent criteria according to the IEC 61,869 series, such as high accuracy in terms of voltage, current, and frequency measurements [7]. Also, LPIT applied in medium voltage aerial conductors must attend practical criteria adopted by Utilities companies [4,5,6], such as, be easy to install and uninstall without disconnecting the distribution network. Once the interruption in service is not desirable when installing temporary metering equipment, the distribution network cannot be switched-off to connect the sensor. A fiberglass hot stick and aerial lifts are normally used to connect the sensor in the conductor, protecting the electric utility workers from electric shock. Other parameters that cannot influence the sensor accuracy are external temperature, the electrical and magnetic fields in the vicinity of the sensors, the aeolian vibration, and the wind loading in the conductors. Besides, the sensor must have the conductor-clamping feature for various diameter cables to measure the current precisely.

Many types of stand-alone LPIT have been developed for many applications and using different technologies. LPIT is a combination of sensors, signal-processing unit, or an intelligent electronic device (IED) and communication interface. Standard LPIT uses Rogowski coils or current transformers for current sensing and capacitive, resistive, or capacitive-resistive dividers for voltage sensing [8]. The signal-processing unit transforms the analog signal of current and voltage into a digital signal. In a standard LPIT, the communication interface is equipped with radiofrequency or cellular communications capabilities [9]. One disadvantage of standard LPIT is the absence of galvanic insulation between the LPIT itself and the signal-processing unit when the signal-processing unit is separated from the sensors. In this case, a copper wire makes the connection between the sensor and the signal-processing unit. When overvoltage events occur (e.g., lightning strikes), they can damage the LPIT.

There are also LPIT where the signal-processing unit is integrated with the sensors and the signal transmission is performed directly from the medium voltage by using the radio frequency or cellular communications capabilities. In this case, the medium voltage conductor itself inductively powers the sensor and the processing unit, however, there are limitations for this application when the conductor current is below few Amperes [10,11].

The second type of LPIT, particularly well developed for high voltage systems, is based on optical fibers technology. Sensors based on optical fiber technology are very suitable for use in power grids due to their excellent characteristics, such as high electrical insulation, non-magnetic saturation, oil-free, no risk of explosive failures, and high bandwidth. In these applications, Faraday Effect, Pockels cells, or fiber Bragg grating (FBG) are used to measure current and voltage [12,13,14,15,16,17,18,19]. Also, these pure optical LPIT has moved to medium voltage applications [15,16,17], where the cost challenges are more evident. Table 1 presents the main characteristics of some optical LPIT cited previously. Typical drawbacks for medium voltage optical LPIT are the necessity of complex schemes for temperature and vibration compensation.

The third type of LPIT, the hybrid LPIT, integrates standard electric and electronic sensors, optoelectronic devices, and optical fibers. The optical fibers of hybrid LPIT provide high electrical insulation from the medium voltage and can suppress the problems of lightning strikes damages and the necessity of complex schemes for temperature and vibration compensation.

In [20,21] are presented hybrids LPIT to be applied in medium voltage. These LPITs are composed of a high-impedance resistive/capacitive voltage divider, an analog electronic insulation interface, and an optical fiber. These LPITs, transmit their signals by optical fibers in an analog way and the non-linearity of the optoelectronic transmitter can alter the shape of the measured voltage signal.

A hybrid digital LPIT can improve the sensor signal transmission and maintain the benefits of electrical insulation provided by the optical fibers. An important question addressed here is how to obtain continuous energy to supply a digital LPIT in many different applications.

An interesting technique that can be used to power sensors and electronic devices in a harsh environment is power-over-fiber (PoF) [22]. In a medium voltage LPIT using PoF, the transmission of the signals from the sensor (placed in the conductor) to the processing unit (placed on the ground), and the optical supply power to the sensor, are made by optical fibers. The voltage and current can be monitored using mature electric devices and processed by digital electronic circuits. The use of optical fibers in the section between the medium voltage and ground is very advantageous since optical fibers are made of non-conductive material. This characteristic is relevant for sensors, which are placed in medium voltage conductors, avoiding current leaks. This non-conductive characteristic is especially important when lightning strikes occur in transmission lines and can cause severe damages in non-insulated electronic equipment, as we commented before. Considering that the sensor using PoF has complete galvanic isolation to the ground potential, it is practically immune to lightning effects. Moreover, optical fibers are not affected by electromagnetic interferences, and they can transmit signals without quality degradation. Besides the advantages of optical fiber previously commented on, the employment of a PoF scheme can eliminate the energy supplied by metallic cable, solar panels, batteries, or harvesting devices eventually placed in the sensor, improving the reliability and the security of the system. Also, the effect of vibration and thus the state of polarization change in the optical cables is eliminated as the PoF is not affected by polarization changes, as can occur in polarization-based optical sensors [23]. Another advantage of the PoF technique is that the analog signals from the current and voltage sensors can be converted to digital form in the electronic board of sensors and robustly transmitted to the processing unit. PoF has been described in the literature for general applications [22,24,25]. In power applications, PoF has been used in current, voltage, and temperature sensors [26,27,28,29,30]. Table 2 presents the main characteristics of some hybrid LPIT including the ones using PoF.

In this work, we present the design, laboratory tests, and the field trial results of a power-over-fiber (PoF) low power instrument transformer (LPIT) for voltage and current measurements in the medium voltage. The design of this power-over-fiber LPIT aims to solve the drawbacks presented by the previous technologies, such as the continuous operation (measuring and data transmission) for a wide current range conducted in the medium voltage transmission lines, damage due to lightning strikes, and accuracy dependency on aeolian vibration, position, and temperatures. This paper is organized as follows. Section 2 describes the physical design of the LPIT for the sensor and processing unit. Section 3 describes the laboratory test in the LPIT applying the standards IEC 61869-10 [31] and IEC 61869-11 [32]. Section 4 describes the field evaluation tests of the LPIT that were made in two different conditions: in a standard distribution network and an experimental hybrid fiber/power distribution network. Finally, Section 5 presents the Conclusion.

## 2. The Proposed Power-over-Fiber LPIT

Figure 1a shows a picture of a standard installation of the PoF based sensor (in just one phase to simplify) in the medium voltage conductor. Three sensors are needed to measure the three-phase of the distribution line. Note that in addition to the optical cable, a ground or reference wire connecting the sensor to the ground or neutral wire is necessary to provide improved accuracy for the voltage measurements. This ground or neutral connection is commonly used for any resistive or capacitive dividers. The ground wire is connected to the bottom of the insulator in such a way there are no problems caused by lightning. A diagram of the PoF LPIT is shown in Figure 1b. In the bottom, it is shown the processing unit that is composed of one high power laser (PoF laser) together with the optical reception unit (Rx Data and RX Clock), which receives signals from the remote sensor unit, shown on the top of Figure 1b. Three optical fibers connect the processing unit to the sensor. In the sensor unit, a photovoltaic converter (PVC) receives the power transmitted by the PoF laser. The electrical energy produced by the photovoltaic converter is used to powering up a low threshold laser, electronic circuits, and the sensors (current, voltage, and temperature) used in this unit. The LPIT is composed mainly of two parts: the sensor and the processing units. The sensor unit connected to the medium voltage conductor, measures voltage, current, and temperature. The sensor unit is used to acquires, digitalizes, and sends data to the processing unit. The processing unit is used to power the sensor unit, receive, process, and analyze the voltage, current, and temperature data from the sensor. In a standard application, the processing unit is installed inside an enclosure at the bottom of the pole and, the data are transmitted to a control room by using general packet radio services (GPRS). In the advanced application (see Section 4), the processing unit can be installed for example, in a control room inside a substation and, the signal can be transmitted to the control room using optical fibers of the distribution network.

### 2.1. Sensor Unit

Figure 2a shows the proposed sensor. The sensor is composed of the following parts: head, capacitor, and insulator. The sensor head contains all the electronics for measurements and signal processing, clamp mechanism, structural support, and a shelter. The insulator is used to insulate the medium voltage to the ground potential and at the same time, it has a structural capacitor (C2) which forms a capacitive voltage divider with the capacitor C1 placed on the electronic board of the sensor unit. The sensor unit was designed to allow installation using a regular clamp stick, the sensor head has an eye bolt used to clamp, transport, and attached the sensor unit to the medium voltage conductor. The optical and the ground cables are connected to the sensor unit at the bottom of the insulator.

Figure 3 shows the functional block diagram of the sensor unit. A PVC and voltages regulators perform the sensor power supply. The PVC converts the optical energy into electrical energy and voltage regulators are used to set up all voltage levels used in the sensor.

The powering and the communication between the sensor and processing unit are performed using three 62.5/125 μm multimode optical fibers, which ensure the high electrical insulation between the medium voltage and ground. The power path is composed of one high power laser model TY976 from Skyera (PoF Laser), placed in the processing unit and operating in the wavelength of 976 nm with a typical output power of 2.0 W, an optical fiber rugged cable, and one Si-based PVC model YCH-L200 PPC, from MH GoPower, placed in the sensor unit. The data path (data and clock) is performed by two low-power and low-cost vertical-cavity surface-emitting lasers (VCSEL) model OPV314YAT from Optek/TT Electronics, which are placed in the sensor unit, and operating in the wavelength of 850 nm with typical 1.0 mW output power. One laser is used for clock transmission and, the other one is used for the data signal. The microcontroller driver each laser directly using a polarization resistor.

Some technologies for current sensors were studied in this project, such as magneto-resistance, shunt resistor, Hall sensors, fluxgate, Rogowski coil, and current transformer (CT). The current sensor based on magnetic resistors reports assembly difficulty due to the nature of the resistance variation as a function of the magnetic field, it is necessary to polarize the magnetic resistor for a linear response of the sensor [33]. The shunt resistor is not a good approach due to the necessity to open the line transmission circuit to install the current sensor. Hall Effect [34] and fluxgate [35] based sensors show assembly issues to make the current sensor immune to external interference. Also, power consumption for sensors based on Flux Gate is quite high due to the current consumed in the sensor feedback. Power consumption is also a limiting factor for the Rogowski coil-based sensor due to the use of the integrator circuit necessary for the phase adjustment of the current signal. The current transformer was the technology chosen for the current sensor. The current transformer shows a suitable accuracy, low cost, and mainly the lowest power consumption option.

The power consumption of the sensor is very critical for PoF applications. A CT consists of a ferromagnetic core, a coil, and a burden resistor. The current transformer involves the conductor where the current is flowing. This approach was chosen for this application due to its non-intrusive measurement method, avoiding that the medium voltage conductor is opened for the installation and removal of the sensor. Figure 2b shows the CT installed in the sensor head. To measure different ranges of currents ferrite or nano-crystalline cores can be used in the CT.

The voltage sensor (Figure 2b) is a capacitive series divider composed of the capacitors C1 and C2. C1 is a surface mount device (SMD) capacitor (installed on the electronic board) and C2 is a structural capacitor. In other words, it is fabricated inside the insulator using a specific electrode design [36], which guarantees immunity to electromagnetic interference from external influences. The dielectric material used in the insulator is the cycloaliphatic epoxy resin and the creepage distance depends on the voltage class. It is interesting to comment that the optical fibers pass throw inside the resin insulator. We used optical fibers with a secondary 900 µm tight-buffered coating of thermoplastic material. The average attenuation in the optical fibers induced by the epoxy resin was 0.66 dB in 850 nm, 0.08 dB in 1310 nm, and 0.52 dB in 1550 nm. These attenuations, mainly for 850 nm are completely acceptable for sensor operation.

The medium voltage to be measured is applied in one C1 terminal and the ground reference is applied in one C2 terminal. The value of C1 is chosen to guarantee low voltage at C1 (up to 2 Vpp) and the remaining high voltage is applied to the C2. The voltage measurement is performed across C1. C1 (C0G/NP0 type) shows a suitable temperature coefficient (0 +/− 30 ppm/°C) for high accuracy measurements. Although we have chosen capacitive dividers for voltage sensors, resistive/capacitive or resistive dividers can also be used.

To protect the sensor from lighting damages, the capacitor C1 in the voltage divider (330 nF/50 Vmax) and the operational amplifier coupled with C1 (INA826AIDR/40 Vmax) were designed to support a basic insulation level (BIL) of 400 kV. In this case, the maximum voltage in the operational amplifier input will be 40 V that is the maximum supported by this device. Regarding the insulator structure, the distance between the medium voltage electrode and the ground terminal is 16 mm. The epoxy resin used in the insulator has a breakdown strength of 18 kV/mm. Then the insulator can support BIL of 396 kV. These values of BIL are higher than recommended by the standards.

The sensor unit was designed to be installed on the medium voltage power lines, exposed to the weather conditions. To keep the sensor accuracy over the temperature range, a microelectronic temperature sensor (LM35) was used in the sensor head. It was fixed in the metallic electrode of C2 available in the sensor head was we can see in Figure 2b. In such a way this sensor can measure the temperature of the center of the capacitor C2 once the metallic electrode conducts the temperature to the temperature sensor.

The sensor unit uses an ARM (advanced RISC machine) microcontroller (model MKL25Z128VLK4) to acquire data of voltage, current, and temperature. It calculates the CRC (cyclic redundancy check) and sends the data frame to the processing unit. Voltage and current samples have 16 bits, temperature samples and CRC word have 8 bits. A data frame has a total of 48 bits and it is sent every 40 µs to the processing unit. The sensor sample rate is 25 kHz. The protocol used to send frame data from the sensor to the processing unit is quasi-SPI (serial peripheral interface). A default SPI protocol is implemented using four wires (CLK, CS, MOSI, MISO) for a duplex communication or three-wire (CLK, CS, MOSI) for one-way communication. In this application, the SPI was implemented using two wires clock (CLK) and data (MISO). The SPI protocol enables high-speed communication (10 Mbit/s) suitable for a high sample rate. Figure 4 shows an example of a data frame used in the communication from the sensor unit to the processing unit.

PoF has a limitation in powering high consumption loads. To obtain a low consumption sensor unit, low power electronic components and design techniques were used, as a result, the sensor current achieved was 10 mA and the total power consumption of the sensor unit was 80 mW.

### 2.2. Processing Unit

Figure 5 shows the functional block diagram of the processing unit. For a better understanding, this diagram is limited to phase A. The data and clock signals from the sensor unit are sent to two optical silicon fiber optic receivers model OPF482 from Optek/TT Electronics (Carrollton, TX, USA), placed in the processing unit using two optical fibers of the same rugged cable described before. In the processing unit, there are also a field-programmable gate array (FPGA) and a power quality analyzer model MQ700 from IMS Power Quality.

The FPGA (model XC6SLX9-L1TQG144C) receives the data from the optical receivers, checks the CRC message, adjusts the phase shift, corrects the voltage and current samples due to temperature effect, and sends voltage samples and current samples to the respective external digital to analog converter (DAC). Finally, the signals are amplified and delivered to the power quality analyzer.

The CRC check block analyzes the CRC code of each message received. If the code is incorrect, the entire frame is left. If it is correct, all data are sent to the separator block. In the separator block, there is a counter. It counts the rising edges of the clock signal and detects where each sample starts and ends. The separator block uses a counter to separate voltage, current, and temperature data. The firsts 16 bits (0 to 15) are sent to the phase voltage compensator block, bits from 16 to 31 are sent to the current phase compensator block and, the last 8 bits (32 to 39) are sent to the temperature hysteresis filter.

The phase shift blocks function is used to provide an adjustment of the phase between the voltage and current sensor. These blocks also allow phase compensation in the voltage and current signals due to a capacitive coupling in the power quality analyzer. The phase current shift block also enables a Rogowski coil as a current sensor and in this case, the blocks act as an integrator.

The temperature hysteresis filter ensures that a new temperature value will be sent to the temperature compensation block only when a real 1 °C temperature transition occurs inside the sensor. This block waits for a long string of temperature samples with the same value in the input block to change the temperature value in the output block. The temperature hysteresis filter block avoids transients and unnecessary correction in the voltage and current samples.

The temperature compensation block corrects the voltage and current samples due to the temperature effect in the sensor. The data of temperature, voltage, and current are the inputs, and the voltage and current compensated data are the outputs of the block. The temperature sample is the index of the compensation table, which was experimentally obtained. Voltage and current data are differently affected by the temperature and use different tables for compensation.

Finally, voltage and current samples are sent to an external DAC and delivered to one commercially power quality analyzer MQ 700 from IMS Quality Power (Porto Alegre, Brazil). For future applications, all-digital processing (employed in the FPGA) will be implemented in the power quality analyzer.

## 3. Laboratorial Tests

The performance of the LPIT (Figure 1b) was first evaluated in a laboratory environment, to verify the performance in terms of accuracy and bandwidth, for current and voltage measurements [31,32]. Further, the sensor was submitted to temperature variation and it was evaluated under this condition. Figure 6 shows one set-up used to characterize the sensor in a high-voltage laboratory.

The voltage sensor, as already mentioned in the text, was developed based on a capacitive divider using two series capacitors, a capacitor located on the sensor’s electronic board, and another capacitor being formed by the insulator structure. Capacitors are good voltage-sensing elements, in this application they were chosen mainly due to the low current developed by the measurement circuit and because of their good frequency response.

The ratio error and frequency response of the voltage measurements were evaluated comparatively, that is, the result obtained by the sensor was compared to a known and reliable reference. We used the reference voltage model Fluke 5520A (Everett, WA, USA) for bandwidth measurement, and for voltage measurements, we used the Chroma (Foothill Ranch, CA, USA) 61,504 AC power source, a commercial elevator transformer, and a calibrated voltage divider model Pearson Electronics (Palo Alto, CA, USA) VD-500A. Figure 7 shows the characterization of the ratio error (Figure 7a) and the bandwidth of the voltage sensor (Figure 7b). As shown, the sensor attends the accuracy criteria of IEC 61869-11 [31] being classified as a class 0.5 accuracy voltage sensor for three possible phase-to-phase voltage classes, which are: 11.9 kV, 13.8 kV, and 23.0 kV (these corresponds to 6.87 kV, 7.97 kV, and 13.28 kV phase-to-ground rated voltages, respectively). In Figure 7a, the ratio error is the difference from reference voltage expressed in percentage and the error bars are the standard deviation, also expressed in percentage, obtained after 5 measurements. The squares of Figure 7a delimit the 80% to 120% of each rated phase-to-ground voltage class. Thus, the black square represents the phase-to-ground rated voltage of 6.87 kV, the red square represents the phase-to-ground rated voltage of 7.97 kV and the green square indicates the phase-to-ground rated voltage of 13.28 kV, as according to IEC 61869-11 the sensor must attend the specified class from 80% to 120% of the rated voltage. As can be seen, the same voltage sensor is within class 0.5 accuracy limits for these three different rated voltages. The bandwidth obtained was 3 kHz at 3 dB, that is enough to measure the 50th harmonic voltage. This bandwidth range is related to the sample rate of the sensor unit (25 kHz) and not to the capacitive divider that has a good frequency response.

In the same way as the voltage measurements, the current sensor response evaluation was made comparatively with the results delivered by a known and reliable reference. We used the reference current sensor model Fluke 5520 A and Fluke 5500 Coil. Figure 8a,b show the characterization of ratio error of two sensor versions for two different rated currents, respectively 25 A and 750 A. Figure 8c shows the characterization of the bandwidth of the current sensor. For both current sensors, five measurements were carried out varying the range within 0 to 120% of the rated current. In Figure 8a this variation corresponds to currents ranging from 1.082 A to 26.26 A (or 4.328% and 105.4% of the rated current of 25 A). In Figure 8b this variation corresponds to 25 A to 900 A (or 3.33% to 120% of the rated current of 750 A). The standard deviation of the five measurements was used to obtain the error bars (precision) for each of these percentages of rated currents, while the deviation of the applied current to the mean value of these five measurements is used to calculate the accuracy. These values of error bars and accuracy are expressed in percentage in these graphs. As shown the sensor attends the accuracy criteria of IEC 61869-10 [31] being classified as a class 0.5 current sensor in the case of the rated current of 25 A and class 0.2 for a rated current of 750 A. However, for the rated current of 25 A, three points from 20 to 28% of this rated current are outside this accuracy class, and a lower accuracy class should be selected. In other words, for the lower-rated current of 25 A the correct accuracy class is the 1.0 accuracy class. According to IEC 61869-10, the current sensor must attend the specified class from 5 to 120% of the rated current. The current sensor was developed for a wide measurement range, such as currents from 1.25 to 30 A or 37.5 to 900 A. The ratio error was within the established class for each range, with saturation around 90/900 A (outside the measurement range) using nano-crystalline core, except for the lower range sensor as three points at ~20% rated current were outside the class 0.5 limits. As for the bandwidth, the result obtained was 4 kHz at 3 dB (65th harmonic). For both voltage and current measurements, the sensor unit is available for power quality measurements.

The influence of temperature on the measurements of voltage and current was also verified in the laboratory. In these tests, the sensor unit was exposed to temperature variations in a thermal chamber while voltage and current measurements were performed. Due to experience in previous developments, the sensor was designed with a temperature sensor to measure and transmit this parameter to the processing unit where the temperature effect on the voltage and current parameters were compensated, thus reaching the levels of required accuracy.

For the voltage measurements, the sensor was subjected to a constant voltage of 7.97 kV phase-to-ground, which represents a measurement of about 13.8 kV phase-to-phase. For the current measurements, the applied current value was maintained constant at 100 A (13.33% of rated current). The temperature of the sensor was varied from 0 to 70 °C for both voltage and current measurements. Once characterized (voltage and current as a function of temperature) a correction table was implemented in the FPGA of the processing unit to reduce the sensor’s voltage and current measurement errors (accuracy). Figure 9 and Figure 10 show the measurement before and after implementing the correction for voltage and current measurement, respectively.

As shown in Figure 9, the voltage ratio error was reduced from about 10% to a variation of about 0.32% in the entire operating temperature range. The error reached about 0.32% was below the stipulated 0.5 class limit. In Figure 10, the current ratio error was reduced from about 1.75% to about 0.125%, in the entire operating temperature range. The error reached of about 0.125% was below the 0.2 class limit of this sensor (~0.53% for the 13.33% of the rated current).

After characterizations of ratio error, bandwidth, and temperature compensation the sensor proved to be suitable for short-term laboratory tests (8 h). During the tests, it was noticed that the current was delayed to the voltage in the power quality meter. This problem was due to the coupling capacitors present at the interface between the processing unit and the power quality analyzer. This problem was mitigated through the phase shift blocks present in the FPGA. Through these blocks the current and voltage signals arrive at the same time at the tester, avoiding measurement errors.

The short-term tests were carried out in the laboratory to assess the stability of the measurements and the functioning of the LPIT (Figure 1b). In these tests, a function generator was used to generate the signals corresponding to voltage and current to maintain the stability of these signals at the sensor’s measurement levels. These signals were injected directly into the signal conditioning blocks (Figure 3) bypassing the capacitive divider and the current transformer. The results of the voltage and current measurements are shown in Figure 11.

The function generator provided signals of 450 mV and 5 mV of amplitude to the sensor, these values according to the sensitivity of the sensor correspond to signals of 7974 V and 238 A of voltage and current respectively. As shown in Figure 11, the test was performed for 8 h. During this period, the RMS values of voltage and current were recorded every 10 s. Throughout the test period, the input signals were kept constant.

The maximum voltage variation was 3.94 V and the maximum current variation was 0.17 A, corresponding to an oscillation of 0.049% and 0.071% about the expected measurement. This experiment showed the stability of the complete system, including the PoF powering, acquisition of electrical quantities, data transmission and the data processing. With these results, the LPIT proved to be ready for a real application in a field test.

## 4. Field Tests

The LPIT (Figure 1b) was submitted to the field tests in two application conditions. The first field test was performed in a standard application condition (Figure 12a). In this condition, the processing unit is installed inside an enclosure at the bottom of the sensor pole and, the data are transmitted to a control room by using general packet radio services (GPRS). The low voltage available in the pole was used to powering the processing unit (and consequently the sensor) and the quality power meters.

In the second field test (advanced application) shown in Figure 12b, the processing unit was installed in a control room and, the signal was sent to the control room using optical fibers of a hybrid fiber/power distribution network. Possible use of this option is to supply sensors where there is not low voltage alternate current available. Another possible application is to use the fibers embedded in the conductor to power the sensors, for example in one substation area.

In the field tests, just one phase was used to test the sensors. In the first field test, the sensor was tested as a voltage sensor and in the second test, it was tested as a voltage and current sensor.

### 4.1. Field Test of Standard Sensor Application

The first field test (Standard Application) was carried out in a medium voltage network inside the CPQD Foundation campus. The processing unit and the power quality meter were placed in a metallic enclosure fixed at bottom of the pole. The nominal grid voltage was 11.9 kV resulting in circa 6.87 kV phase-to-ground measurements. In this test, the sensor unit was evaluated as a voltage sensor, and voltage, external temperature, and precipitation data were collected for ten days. The temperature and precipitation values were collected from a weather station close to the sensor installation site. As shown in Figure 13, during the measurement period, no voltage anomaly was detected. The installation details of the sensor can be reached in Appendix B.

Through the temperature variation, the day and night periods can be noticed during the test days. There were also two episodes of rain, as shown in Figure 13. The range of temperature variation was 15.6 °C. The maximum precipitation was 25 mm in this test period. The maximum variation in the measured voltage was 170 V, corresponding to 2.41% of the total variation of the average voltage. The measured values of minimum and maximum voltage were 6.960 kV and 7.043 kV, respectively, a variation of −1.19% and 1.22%. This variation for voltage does not exceed the maximum allowed by Brazilian electricity standards [37].

A correlation plot between voltage and temperature was obtained from the data in Figure 13. The obtained correlation is shown in Figure 14. To quantify the correlation, we make use of the Pearson correlation [38]. Pearson’s correlation coefficient measures the degree of correlation (and whether it is positive or negative) between two variables. According to criteria given in [38], the degree of correlation is classified in:0.9–1: positive or negative indicates a very strong correlation.0.7–0.9: positive or negative indicates a strong correlation.0.5–0.7: positive or negative indicates a moderate correlation.0.3–0.5: positive or negative indicates a weak correlation.0–0.3: positive or negative indicates a negligible correlation.

As can be seen in Figure 14, the Pearson correlation of these data gives a value of −0.15, thus classified as “negligible correlation” between these variables.

### 4.2. Field Test of Advanced Sensor Application

This field test was conducted on the campus of UniverCemig in Sete Lagoas, Brazil. On this campus, there is an experimental hybrid distribution network call Synergic Network (SN). SN is composed of optical phase conductors (OPPC), optical distribution conductors (OPDC), and many other electrical and optical network accessories. Many advanced applications have been tested in this experimental network [39]. We installed a current and voltage sensor that is specified to work up to 13.8 kV/25 A. The installation details of the sensor can be reached in Appendix B and the Appendix A shows the sensor installation.

Before starting the field test, we evaluated the maximum distance possible to powering the sensor unit using optical fibers concatenation of the SN and additional optical fibers spools. In the test, the distance was increased from the sensor to the processing unit using extra optical fiber spoons. The optical power to the sensor unit was measured in the OTB and the signal quality was measured in the processing unit using an oscilloscope. Figure 15 shows the 976 nm optical power level versus the link length. The maximum distance obtained with an acceptable signal quality was 1.8 km operating the PoF laser with 1.2 W. However, considering that the maximum optical power of this laser is 2 W a maximum distance of 2.8 km can be reached.

After the sensors start operating in the SN, we collected some voltage, current, and harmonics data. Figure 16a shows the waveforms of the voltage. Figure 16b shows the current waveform, and Figure 16c shows the harmonics content for these signals. The final distance between the sensor to the processing unit was 687 m using an extra optical fiber spool. All data shown in Figure 16 were obtained from power quality analyzer MQ-700. This module has embedded the requirements of Brazilian electricity standards [37].

Figure 17 shows the voltage and current collected for approximately 3 months. In this test period, the voltage measured by the system had an RMS value of 8 kV, suitable for a 13.8 kV phase-to-phase grid. The maximum and minimum current measurement for the sensor was 0.51 A and 4.23 A suitable for the typical load of the campus. The measured values of minimum and maximum voltage were 7.67 kV and 8.47 kV respectively with a variation of −3.78% and 6.35%. Like the previous test, this variation in voltage measurement does not exceed the maximum allowed by Brazilian electricity standards [37], except for the measurement of maximum voltage that exceeds +1.35% from the limit.

The voltage and current data were obtained for almost 3 months. In this period, the powering system works well demonstrating that the sensor is robust enough for outdoor applications. In many applications for power quality, the sensor is used in a specific point of the electric distribution network for just some days, e.g., one week or ten days.

## 5. Conclusions

In this work, we presented the design, laboratory tests, and the field trial results of a PoF based LPIT for voltage and current measurements in the medium voltage. The design of this power-over-fiber LPIT aims to solve the drawbacks presented by the previous technologies, such as the continuous operation (measuring and data transmission) for a wide current range conducted in the medium voltage transmission lines, damage due to lightning strikes, accuracy dependency on aeolian vibration, position and temperatures.

We presented in detail the design of the LPIT including the sensors used to measure voltage, current, and temperature. We detailed also the technique used to transmit the digital data from the sensor unit to the processing unit, which uses a special synchronism technique. We detailed the optical transmission and powering interfaces in the sensor and the processing unit. The losses of the fiber inside the electrical insulator were reported. In the processing unit, the mechanism of temperature and phase compensation was also presented.

The results of the laboratorial tests of the sensor were reported according to the criteria of IEC 61869-10 and IEC 61869-11. The voltage sensor is classified as class 0.5 for three nominal phase-to-phase voltages: 11.9 kV, 13.8 kV, and 23.0 kV, which corresponds to 6.87 kV, 7.97 kV, and 13.28 kV phase-to-ground values. The current sensor is class 1.0 for 1.25 to 30 A range and class 0.2 for 37.5 to 900 A range. The temperature correction permits the sensors to operate in these classes from 0 to 70 °C.

We also presented the results of field tests of the sensor made in two different conditions. The first field lasted 10 days test and it was performed in a standard application condition, i.e., the sensor was installed on the conductor of a medium voltage line and the processing unit of the sensor was installed on the bottom of a pole. The data of the processing unit was transmitted to a control room by using general packet radio services (GPRS). In this case, the processing unit was powered using the low voltage alternate current available in the pole. In this first test, we obtained the correlation between voltage and temperature using the Pearson correlation that was classified as “negligible correlation” between these variables.

In the second field test (advanced application), the processing unit was installed in a control room and, the signal of the sensors, which were installed remotely (687 m from the processing unit), was sent to the control room using multimode optical fibers of an experimental hybrid fiber/power conductor. The optical powering was performed by other multimode fiber available in this conductor. The voltage and current data obtained during almost 3 months in this field test demonstrated that the sensor is robust enough for outdoor applications.

Regarding the advanced application, the installation of optical cables to connect the sensing head and the processing unit is easier in a substation than in the public areas, where the medium voltage distribution network is commonly installed. However, this application could be available in future developments of hybrid PoF sensors that use ultra-low-power devices and single-mode fibers, once single-mode fibers have been used on a large scale nowadays for both telecommunications and Utilities companies.

Also, we presented the installation aspects of the sensor demonstrating that it attends to the practical criteria adopted by Utilities companies including voltage measurements without removing the coating of the covered conductors.

Finally, we can compare the results obtained with the proposed LPIT with the other proposed solutions published in the literature. The proposed solution is the first demonstration of a voltage and current sensor for medium voltage working together in the same case and using PoF. Also, the sensor demonstrated the simultaneous measuring and data transmission of voltage and current in a continuous operation for low current levels (<1 A), conducted in a 13.8 kV medium voltage transmission line. In terms of accuracy, the laboratory results demonstrated that the sensor exceeds the proposed solutions detailed in Table 2 for other PoF or hybrid sensors.

We believe that these studies aim to incorporate optical fiber and devices, digital technologies, communications systems in electrical systems driving their evolution.

## Figures and Tables

**Figure 1 sensors-21-00547-f001:**
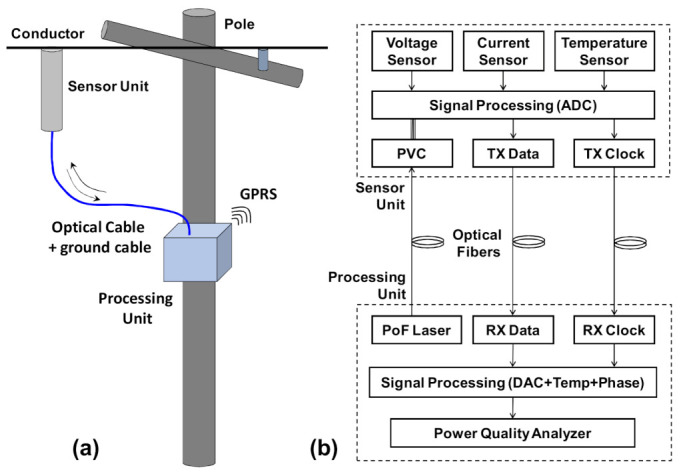
Power-over-fiber voltage and current LPIT. (**a**) Standard installation of the LPIT and (**b**) block diagram of the LPIT.

**Figure 2 sensors-21-00547-f002:**
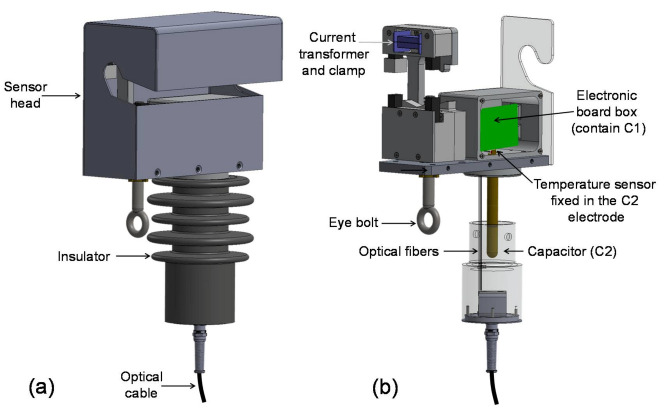
(**a**) The main parts of the sensor unit and (**b**) sensor head without the protective cover showing: current transformer, clamp, eye bold, electronic board box and the optical fibers and the capacitor C2 shown without the insulator layer.

**Figure 3 sensors-21-00547-f003:**
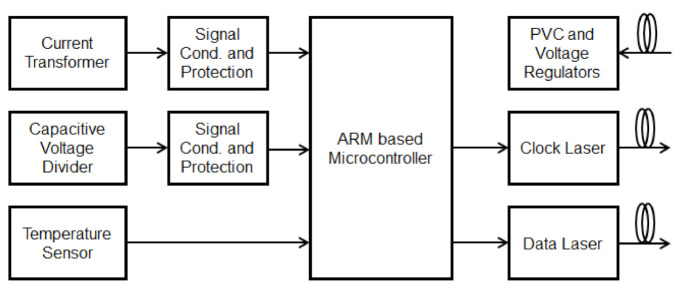
Block diagram of the sensor unit.

**Figure 4 sensors-21-00547-f004:**
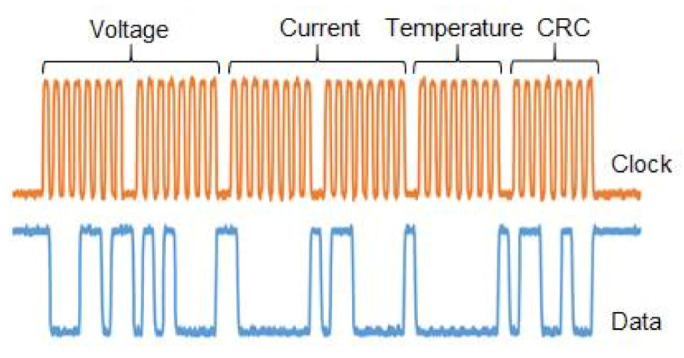
An example of a data frame used in the communication from the sensor unit to the processing unit.

**Figure 5 sensors-21-00547-f005:**
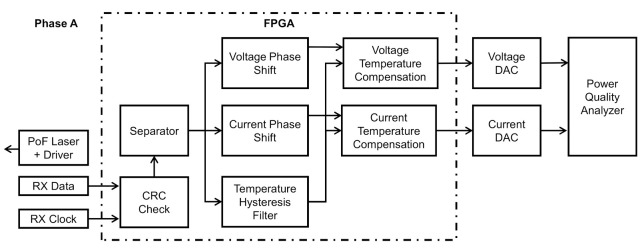
Processing unit block diagram.

**Figure 6 sensors-21-00547-f006:**
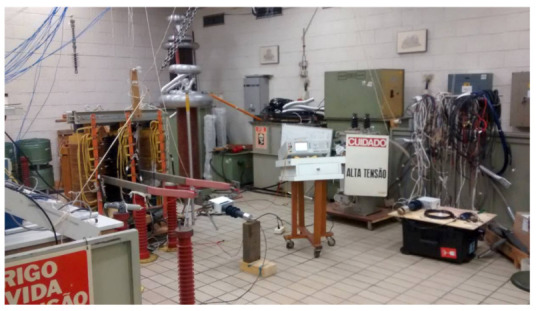
Set-up used to characterize the sensor in a high-voltage laboratory.

**Figure 7 sensors-21-00547-f007:**
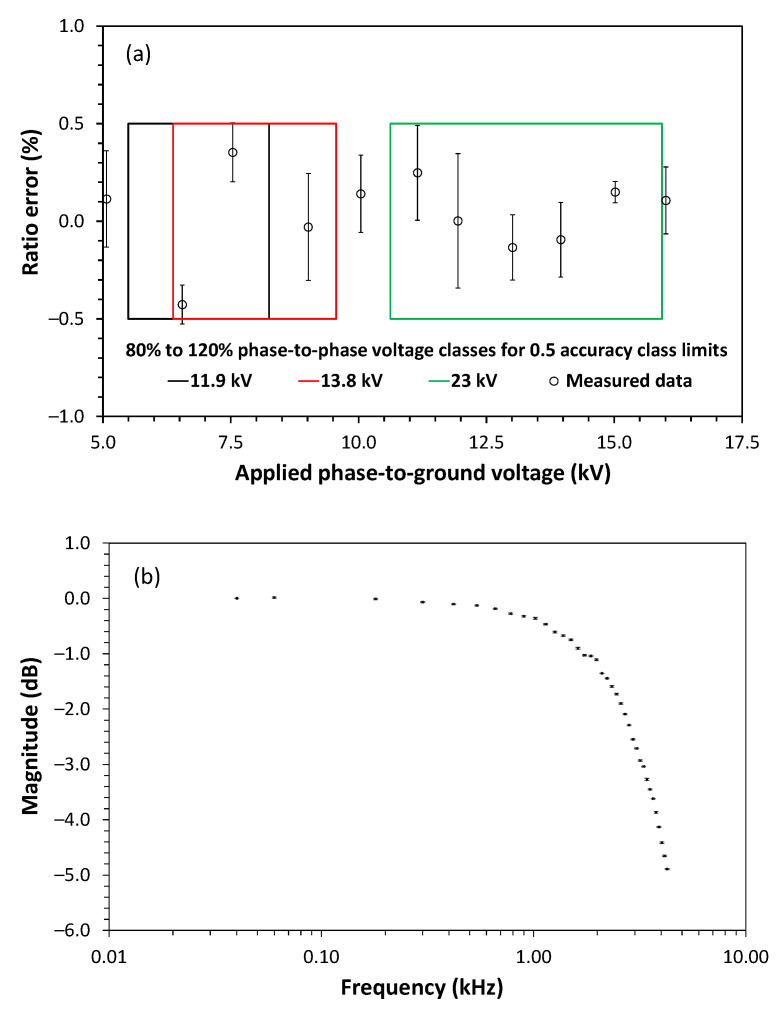
Voltage sensor characterization: (**a**) ratio error and (**b**) bandwidth.

**Figure 8 sensors-21-00547-f008:**
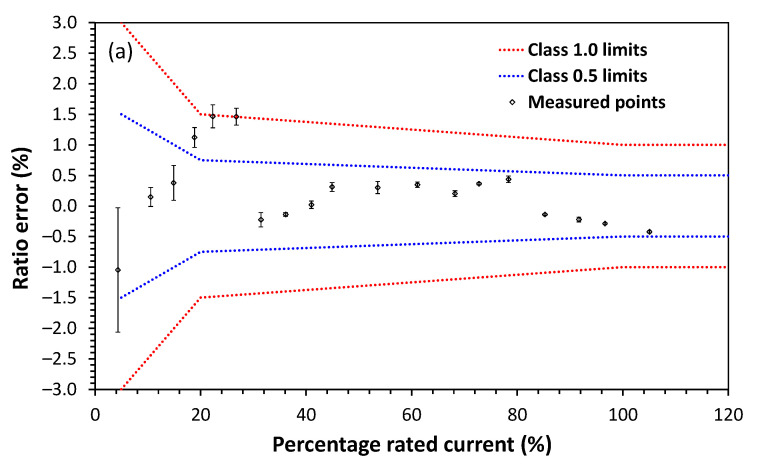
Current sensor characterization: (**a**) ratio error for 1.25–30 A range for the rated current of 25 A, (**b**) ratio error for 37.5–900 A range for the rated current of 750 A and (**c**) bandwidth.

**Figure 9 sensors-21-00547-f009:**
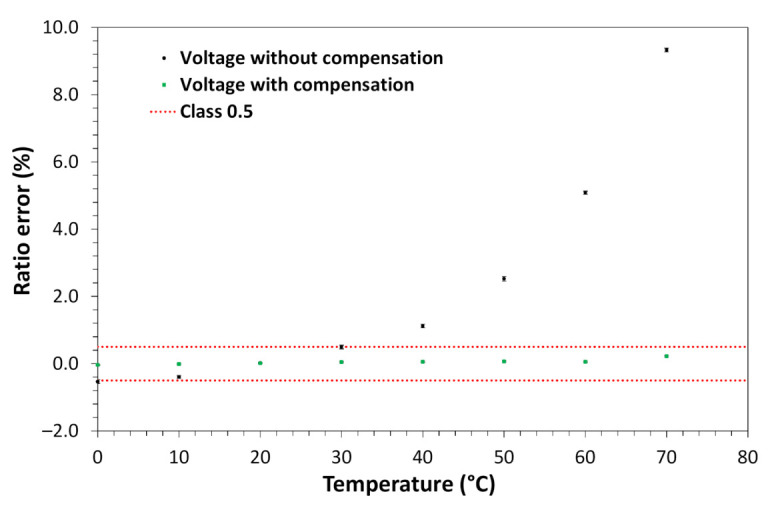
Voltage sensor measurements with and without temperature compensation.

**Figure 10 sensors-21-00547-f010:**
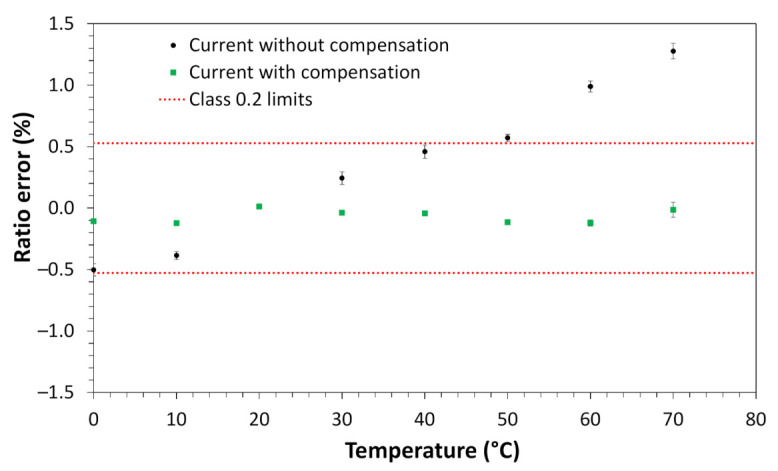
Current sensor measurements with and without temperature compensation.

**Figure 11 sensors-21-00547-f011:**
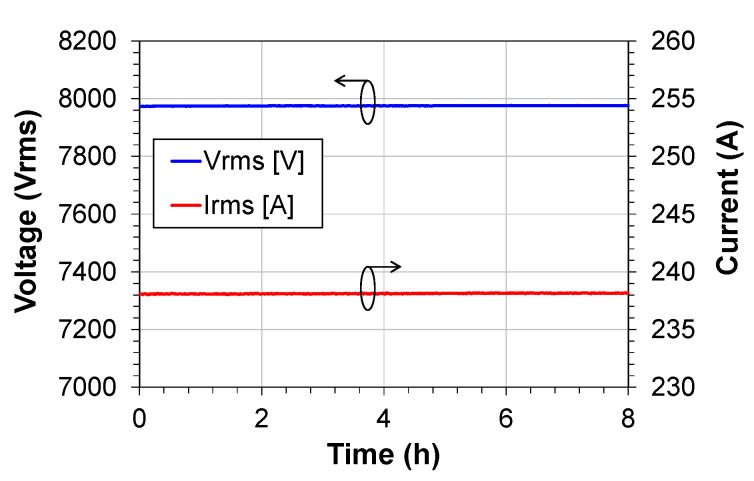
Results of the voltage and current measurements in the short-term test.

**Figure 12 sensors-21-00547-f012:**
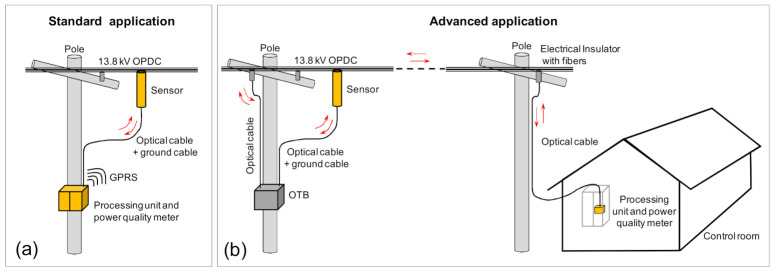
The two applications that were tested in the field for the sensing system. (**a**) Standard application and (**b**) advanced application.

**Figure 13 sensors-21-00547-f013:**
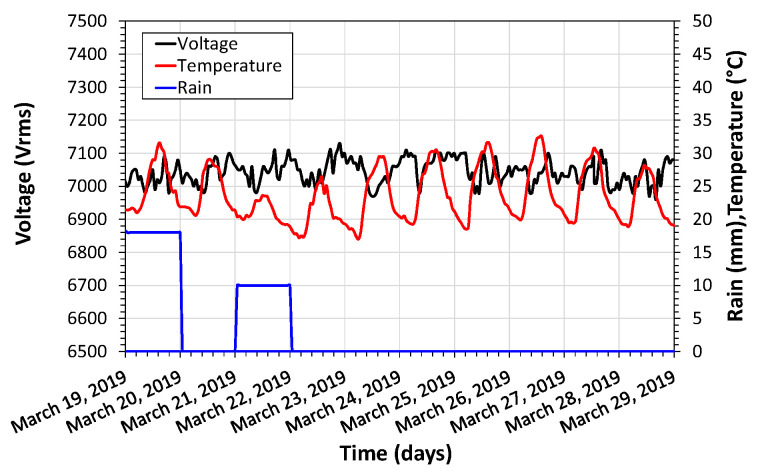
Voltage, temperature variation, and precipitation measurements during the field-test of the sensor in a standard application.

**Figure 14 sensors-21-00547-f014:**
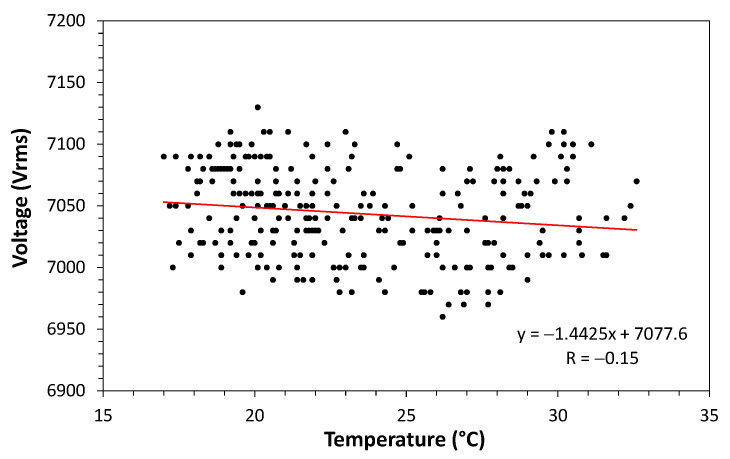
Correlation plot between voltage and temperature obtained from the data of Figure 13.

**Figure 15 sensors-21-00547-f015:**
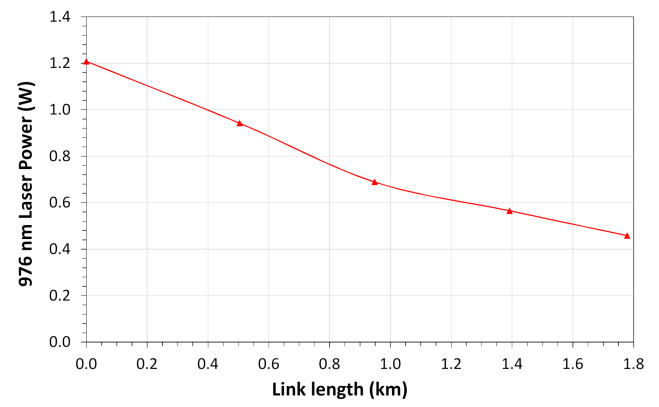
Optical power level to sensor supply versus the link distance.

**Figure 16 sensors-21-00547-f016:**
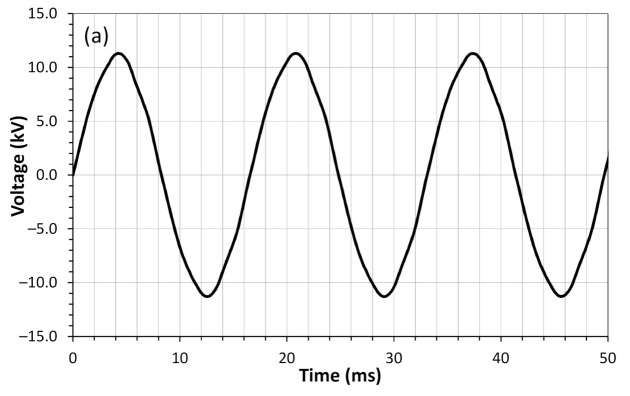
(**a**) Waveforms of the voltage, (**b**) current, and (**c**) the harmonics content for these signals.

**Figure 17 sensors-21-00547-f017:**
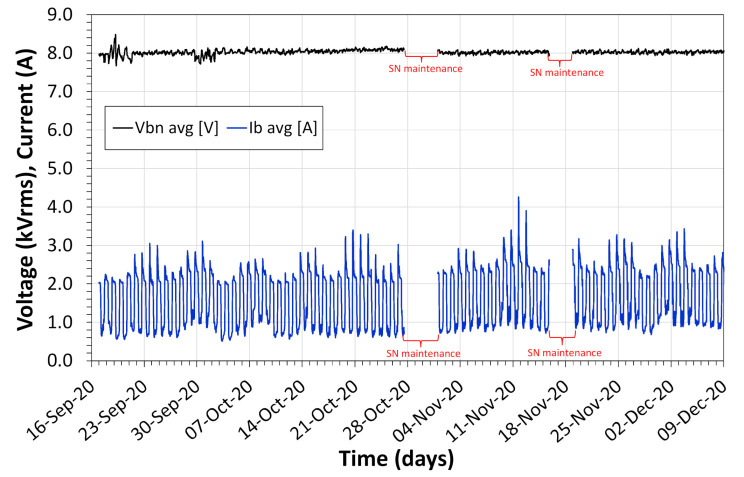
Voltage and current data obtained during almost 3 months in the field test of the sensor in the advanced application.

**Table 1 sensors-21-00547-t001:** Characteristics of optical LPIT described in the literature.

Ref.	Technology	Technical Characteristics	Comments
[4] ^1^	Polarization	Voltage: 16 kV, Accuracy 0.2%	Sensor mounts to the conductor. An accuracy correction factor for phase angle may be required. Harmonic readings are very dependent on frequency. Calibration curve very nonlinear. Installation environmental concerns: aeolian vibration, ice buildup, and wind loading of conductors.
[16] ^2^	Polarization	Voltage: 25 kV, Accuracy: 0.3%	The light source is located at the high voltage side. A co-located impedance string feeds the integrated power supply to energize the light source at the high voltage end, and a second power supply at the neutral end to power the optical receiver electronics and digitizer. The hollow insulator is pressurized to 15 psi with dry nitrogen gas.
[18]	FBG	Voltage: 11 kV, Accuracy 2%	The sensor prototype, comprising a piezoelectric transducer and a fiber Bragg grating (FBG). Temperature tests and field trials were not performed.
[19]	Polarization	Temperature cycling 5000 cycles, −25 °C and 65 °C over several years), constant high temperature at dry conditions up to 115 °C and damp heat 85% Rh at 65 °C and 85 °C	It was presented a study of long-term reliability of interferometric fiber-optic current sensors designed for high voltage (HV) application. Failures in the main devices of the sensor system were related during the tests. A field trial of a three-phase sensor system integrated into 420 kV circuit breakers was carried out over several years.

^1^ OVCS, ^2^ OVT.

**Table 2 sensors-21-00547-t002:** Characteristics of hybrid LPIT described in the literature.

Ref.	Technology	Technical Characteristics	Comments
[4,16] ^3^	Self-powered	Current: 0.2–10 kA, Accuracy: 0.3% Voltage: 25 kV, Accuracy: 0.3%	An accuracy correction factor for phase angle is required. The sensor may allow harmonic readings. Readings are frequency dependent. Calibration curve nonlinear. Conductor hold-down clamps may be susceptible to Aeolian vibration. A covered conductor must be stripped back to allow voltage potential contact. The gas-filled cavity of the sensor may be a maintenance issue.
[26]	PoF	Current: 1–200 A, Accuracy ±1%	Uses a commercial Rogowski coil as a current sensor. PPC voltage and sensor temperature are measured to improve sensor accuracy. Uses a dedicated pulse position modulation (PPM) to data transmission. The Rogowski coil is not a split-coil and presents difficulty in installation in the legacy network.
[27]	PoF	Current: 4000 A, Accuracy ±0.5%	The current sensor is employed by a shunt resistor and power and data transmission are performed by optical fibers. Shunt resistor represents an installation drawback because requires insertion into the circuit under measurement.
[28]	Battery	Current: 0–800 A, Accuracy 2% Voltage: 0–20 kV, Accuracy 5%	Resistor divider for voltage measurement and current clamp (CT with air gap) for current measurement. Current measurement susceptible to nonlinearity due to interference from adjacent magnetic fields. A covered conductor must be stripped back to allow voltage potential contact.
[29]	PoF	Current: 100–650 A, Temp.: 10–90 °C	Current and temperature measurement for sag estimation is proposed. RMS current is calculated at the sensor and transmitted to the ground side system. Harmonic readings are not reported.
[30]	PoF	Current: 20–480 A, Accuracy 0.5%	Error drifts 1% over −20 to 80 °C, narrow band in frequency response: 20 Hz to 250 Hz.

^3^ MetPod.

## Data Availability

Data is contained within the article.

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
