# Peer review of "Power-over-Fiber LPIT for Voltage and Current Measurements in the Medium Voltage Distribution Networks"

_sensors, 2021, doi:10.3390/s21020547_

Round 1
Reviewer 1 Report
The manuscript is interesting. It has been improved. However, some additional improvements should be applied.
1) The title seems to be incomplete. I think that at the end of the title the words “Distribution Networks” or “Power Lines” should be added.
2) The abstract is too long and contains too many details.
3) The caption of Figure 6 is: “Voltage sensor characterization: (a) linearity and (b) bandwidth.” Where are parts “(a)” and “(b)”?
Reviewer 2 Report
The contents of the manuscript are well described to be basically understandable. However, the reviewer believes the following revisions will make the manuscript better.
(1) The Abstract and the last paragraph of the Introduction section are almost identical to the Discussion/conclusion section, which causes redundancy. The Abstract should be more concise. Similarly, detailed explanations of the work are not necessary in the Intrduction section and thus its last paragraph should be more concise, too.
(2) "5.Discussion/Conclusion" section should be simply named as "5.Conclusion".
(3) In the figure caption of Fig.8, in order to avoid confusions of readers, it should clearly expressed, for example, as "(a) ratio error for 1.25-30 A range for the rated current of 25A, (b) ratio error for 37.5-900 A range for the rated current of 750A".
(4) On page 7, line 229, "to become the current sensor immune" should be "to make the current sensor immune".
(5) On page 15, line 423, "in Figure 10" should be "in Figure 11". (Please check)
Reviewer 3 Report
Dear Authors,
Regarding the first round review of this manuscript, the reviewer has the following comments:
- please add the block-diagram due to the proposed work in the introduction and explain about it in one paragraph. To following your manuscript more easily please in the other sections refer to the proposed method's block-diagram.
- In the last paragraph of introduction, the authors needs to add paper introduction!!!
- How you can validate the stability and reliability in your method?
- In Figure 4, how the authors find the number of samples for voltage, current,...
- Do you used Microcontroller or FPGA? the reviewer confused about it. please explain more clear.
- Figure 8 needs more explanation.
Please answer all questions in the response letter and prepared the face of manuscript after major revisions.
Regards,
Round 2
Reviewer 3 Report
Dear Authors,
Thank you for your cover (response) letter. Regarding the second round review of this manuscript, this manuscript can be accept for further processing.
Regards,
Author Response
Dear Reviewer,
Thank you for your information.